# Serological Response After the Fourth Dose of COVID-19 Vaccine in Highly Immunosuppressed Patients

**DOI:** 10.3390/vaccines13100994

**Published:** 2025-09-23

**Authors:** Abelardo Claudio Fernández Chávez, Paula Navarro López, Ana De Andrés Martín, Daniel Leonardo Sánchez Carmona, Guillermo Yovany Ordoñez León, Jesús María Aranaz Andrés

**Affiliations:** 1Department of Preventive Medicine and Public Health, Ramón y Cajal University Hospital, IRYCIS (Ramón y Cajal Institute for Health Research), 28034 Madrid, Spain; abelardoclaudio.fernandez@salud.madrid.org (A.C.F.C.); jesusmaria.aranaz@salud.madrid.org (J.M.A.A.); 2Department of Immunology, Ramón y Cajal University Hospital, 28034 Madrid, Spain; aandresm@salud.madrid.org; 3Department of Preventive Medicine, University Clinical Hospital, 47003 Valladolid, Spain; danieleosanchezc@gmail.com; 4Department of Preventive Medicine, Gregorio Marañón University Hospital, 28007 Madrid, Spain; guillermo.ordonez@salud.madrid.org; 5Servicio de Medicina Preventiva y Salud Pública, Ramón y Cajal University Hospital, IRYCIS, CIBER de Epidemiología y Salud Pública (CIBERESP), 28034 Madrid, Spain; 6Facultad de Ciencias de la Salud, Universidad Internacional de La Rioja (UNIR), 26006 Logroño, Spain

**Keywords:** COVID-19, immunosuppression, vaccination

## Abstract

Introduction (Objectives): This study aimed to evaluate the serological response to a fourth dose of mRNA COVID-19 vaccine in patients with conditions that confer a high risk of severe disease, particularly those with high-level immunosuppression. Methods: An observational study was conducted at the Ramón y Cajal University Hospital between February and August 2022. Adults (≥18 years) with high-risk conditions who had received four doses of either BNT162b2 or mRNA-1273 were included. Anti-spike IgG levels were measured ≥14 days post-vaccination. An adequate response was defined as an antibody concentration ≥260 BAU/mL. Results: A total of 943 patients were analyzed; 846 (89.7%) achieved an adequate response. In the bivariate analysis, patients aged 60–74 years had a higher risk of inadequate response compared to those aged 18–39 years (OR 1.824 vs. OR 0.257). Female sex was associated with a higher risk of inadequate response (OR 1.522; 95% CI: 0.974–2.371). In multivariable logistic regression, patients with high immunosuppression had a higher, though not statistically significant, risk of inadequate response compared with those without. Discussion: Our findings are consistent with international evidence suggesting that age and certain clinical factors reduce vaccine immunogenicity. The observed paradoxical effect of sex could reflect the higher prevalence of aggressive immunosuppressive therapies among women in the study cohort. Conclusions: Most immunosuppressed patients achieved seroconversion after the fourth dose. These results underscore the need for tailored vaccination strategies and additional measures in highly immunosuppressed subgroups.

## 1. Introduction

In February 2021, Spanish health authorities recommended prioritizing vaccination for patients at risk of severe disease from SARS-CoV-2 infection: solid organ transplant recipients, hematopoietic stem cell transplant (HSCT) recipients, patients with chronic kidney disease on hemodialysis, patients with combined primary immunodeficiencies, oncology patients undergoing chemo- or radiotherapy, individuals with HIV and CD4 < 200 cells/μL, patients with Down syndrome, and those receiving immunosuppressive or immunomodulatory therapies [1,2,3].

Scientific evidence indicates that within this risk group, individuals with specific underlying conditions exhibit a reduced capacity to mount an adequate humoral immune response to vaccination, thereby constituting a distinct subgroup of patients with “pathological conditions that induce high-level immunosuppression” [3]. This underscores the need to quantify the humoral immune response following COVID-19 vaccination. However, studies show that serological testing has important limitations: routine serology does not measure neutralizing antibodies nor distinguish the target strains of the antibodies. [4,5].

Studies classify recipients of solid organ or hematopoietic stem cell transplants, patients with hematological malignancies, oncology patients receiving chemotherapy, patients on immuno-targeted therapy, and those with combined primary immunodeficiencies as having “pathological conditions that induce high-level immunosuppression”. [4,6,7]. This subgroup does not include patients on hemodialysis, those receiving non-targeted immunosuppressive therapy, or individuals with Down syndrome, cystic fibrosis, or HIV with CD4 < 200 cells/μL, who are collectively referred to as patients without high-level immunosuppression [8,9].

Publications indicate that, within the high-level immunosuppression subgroup, COVID-19 vaccination does not guarantee the expected immune response, even among patients who exhibit an apparently adequate humoral response [10]. A study by H-Vasquez et al. observed that the cellular immune response is limited in older adults and in patients with hematologic malignancies, even prior to vaccination [11]. This finding has prompted the search for alternatives to vaccination, such as passive immunization through pre-exposure prophylaxis, to provide protection against potential SARS-CoV-2 infection [12].

The main objective of this study is to describe and analyze the serological response after the fourth dose of the COVID-19 vaccine in highly immunosuppressed patients.

## 2. Materials and Methods

This cross-sectional observational study was conducted between 15 February and 31 August 2022, and included patients aged ≥18 years with underlying conditions associated with an increased risk of severe COVID-19 who had received four doses of an mRNA vaccine (BNT162b2 or mRNA-1273) at the Preventive Medicine Department of Ramón y Cajal University Hospital. Patients with documented hypersensitivity or severe allergic reactions to any component of the mRNA vaccines were excluded.

High-level immunosuppression was defined according to specific clinical conditions: solid organ or hematopoietic stem cell transplant recipients ≥3 months post-transplant; patients with hematologic malignancies; patients with primary immunodeficiencies classified as severe combined immunodeficiency; cancer patients receiving active cytotoxic chemotherapy; and patients on targeted immunotherapies, specifically anti-CD20 or anti-TNFα agents (e.g., adalimumab, infliximab).

Antibody concentrations were measured ≥14 days after the fourth vaccine dose using a standardized quantitative immunosorbent assay (ELISA/CLIA), expressed in BAU/mL according to the WHO international standard. An adequate serological response was defined as an anti–SARS-CoV-2 spike IgG antibody concentration ≥260 binding antibody units per milliliter (BAU/mL), a threshold associated with vaccine-induced protection in previous studies, as values above 250–300 BAU/mL correlate with significant protective immunity. A concentration < 260 BAU/mL was considered an inadequate serological response [13]. This threshold has been applied in immunosuppressed populations to identify suboptimal responders and to guide additional prophylactic interventions, such as monoclonal antibodies [14]. Data were obtained from the Preventive Medicine Department vaccine registry and included clinical-epidemiological variables such as age, sex, comorbidities, serological results, vaccination dates, and the interval between vaccination and serology testing.

Categorical variables were summarized as counts and percentages, and continuous variables as median and interquartile range (IQR). Group comparisons (adequate vs. inadequate humoral response) were performed using the chi-square test or Fisher’s exact test, as appropriate, for categorical variables, and the two-sided Wilcoxon rank-sum test for continuous variables. Associations were assessed with univariable logistic regression, reporting odds ratios (OR) with 95% confidence intervals (CI).

Bivariate analyses compared clinical and epidemiological characteristics between patients with adequate and inadequate serological responses. Multivariable logistic regression was then used to assess the association between serological response (coded as 1 = inadequate, 0 = adequate) and immunosuppression status (coded as 1 = high-level; 0 = without-high-level), adjusting for age and sex. Odds ratios with 95% confidence intervals were calculated.

Statistical analyses were performed using Stata version 16. This was an observational, analytical, cross-sectional study designed to evaluate the association between exposure variables and serological response in immunocompromised patients.

## 3. Results

A sample of 943 patients (370 females) with risk factors for COVID-19 who had received four vaccine doses at the Preventive Medicine Department of Ramón y Cajal University Hospital was evaluated; 846 individuals (89.7%) achieved an adequate serological response. Among female patients, 87.3% (323/370) achieved an adequate response, compared with 91.2% (523/573) of male patients.

Descriptive comparisons showed that a greater proportion of inadequate responders were aged ≥60 years (Table 1). Vaccine response was similar between patients with and without high-level immunosuppression. The most frequent conditions were solid organ transplantation (*n* = 446), targeted immunosuppressive therapy (*n* = 127), and hematopoietic stem-cell transplantation (*n* = 118); none showed statistically significant differences between adequate and inadequate responders. In the chemotherapy subgroup, only 2 of 75 patients (2.7%) showed an inadequate response, and no inadequate responses were observed among patients with HIV infection or primary immunodeficiencies. The interval between the fourth vaccine dose and serological testing was similar between groups (median 18 days; IQR 16–26 vs. 16–23).

In Figure 1, among patients with high-level immunosuppression, those with an inadequate humoral response were generally older than adequate responders (median 65 vs. 60 years; *p* = 0.05), except among patients with solid tumors, in whom ages were similar (median 70 years in both groups; *p* > 0.05). Female patients predominated among those receiving targeted therapy, those with hematologic malignancies, and solid-organ transplant recipients, whereas male patients were more frequent among those with solid tumors and among hematopoietic stem-cell transplant recipients.

In Figure 2, among patients without high-level immunosuppression, no inadequate vaccine response was observed in those undergoing hemodialysis or with primary immunodeficiency. By contrast, within the group receiving non-targeted immunosuppressive therapy, patients with an inadequate humoral response were older than adequate responders (median 72 vs. 58 years; *p* < 0.01).

In univariable logistic regression, age 60–74 years (vs. 18–39 years) was associated with higher odds of an inadequate response (OR = 1.824; 95% CI: 1.193–2.788; *p* = 0.005), while male sex was associated with lower odds (OR = 0.657; 95% CI: 0.421–1.025; *p* = 0.049). Neither age ≥75 years (OR = 1.098; 95% CI: 0.518–2.329; *p* = 0.810) nor high-level immunosuppression (OR = 1.711; 95% CI: 0.612–6.634; *p* = 0.303) reached statistical significance (Table 2).

In multivariable logistic regression, two variables were independently associated with inadequate humoral response to the fourth COVID-19 vaccine dose. Participants aged 60–74 years had over fivefold higher odds of an inadequate response compared with those aged 18–39 years (OR = 5.29; 95% CI: 1.26–22.29; *p* = 0.023), and male sex was associated with a 39% reduction in the odds of an inadequate response compared with female sex (OR = 0.61; 95% CI: 0.40–0.94; *p* = 0.024). Other variables, including high-level immunosuppression (OR = 1.71; 95% CI: 0.60–4.85; *p* = 0.316), age ≥ 75 years (OR = 4.24; 95% CI: 0.86–20.81; *p* = 0.075), and age 40–59 years (OR = 2.92; 95% CI: 0.68–12.58; *p* = 0.149), showed non-significant trends toward higher odds of inadequate response (Table 2).

## 4. Discussion

In this cross-sectional observational study of immunosuppressed patients, we evaluated the serological response to a fourth COVID-19 vaccine dose. Individuals with conditions associated with high-level immunosuppression, including those receiving targeted therapies and solid organ transplant, tended to show lower antibody responses [15]. However, although high-level immunosuppression showed a trend toward worse response (OR > 1), the wide confidence interval reflects limited statistical power to confirm an association. This result should therefore be interpreted with caution and does not imply a statistically significant difference.

Multivariable logistic regression identified age 60–74 years as significantly associated with reduced humoral immunogenicity after a fourth COVID-19 vaccine dose, with more than a fivefold increase in the odds of an inadequate antibody response compared with the youngest adults, consistent with previous reports [16]. The impaired immune response observed in older adults may be attributed to immunosenescence, a multifactorial and dynamic phenomenon that affects both natural and acquired immunity. Immunosenescence plays a key role in the response to vaccination and infections in older people and is characterized by age-related immune decline, including decreased antibody production and reduced B- and T-cell activity [15,17].

Among the patients evaluated, males demonstrated a higher vaccine response compared with females. Sex-based differences in immune responses have been documented in previous research, and several studies suggest that women, on average, develop higher antibody responses than men after vaccination [18]. However, in our study, female patients more frequently had autoimmune diseases requiring intensive immunosuppressive therapy, a factor that may have counteracted the female immunological advantage [19]. This characteristic of our study population provides a plausible biological and clinical explanation for the lower response observed in women, despite the opposite trend described in the general literature.

Evaluating vaccine response in patients with high-level immunosuppression is challenging. In our study, solid organ transplant recipients emerged as a key group with inadequate responses. Multiple factors, including the use of immunosuppressive agents to prevent graft rejection, contribute to their diminished immunity. In subgroup analyses, solid organ transplant recipients had a slightly lower adequate response rate (≈88%) than the overall cohort, whereas solid cancer patients receiving chemotherapy showed unexpectedly high response rates (≈98% with adequate response; adjusted OR 0.22; *p* = 0.02, compared with others). Importantly, no single immunosuppressive therapy was significantly associated with humoral nonresponse [20].

In a cohort study by Thomson et al., renal transplant recipients receiving two or more classes of immunosuppressive drugs had significantly reduced odds of seroconversion after the fourth vaccine dose (OR 0.41; 95% CI 0.17–0.90; *p* = 0.033). Notably, 18.8% of infection-naïve recipients remained seronegative, highlighting their persistent vulnerability despite repeated vaccination [21]. Similarly, Gleeson et al. reported that even patients immunized with two or three doses before transplantation exhibited inadequate vaccine responses, which improved when the fourth dose was administered after a median interval of 226 days. These findings highlight the potential role of monoclonal antibody prophylaxis during the first 3–6 months post-transplant [22].

In cohorts of kidney transplant recipients and onco-hematological patients, high seroconversion rates were observed after the fourth dose (between 76% and 95% responders), although a subgroup persisted without an adequate humoral response [23]. In the United Kingdom, antibody positivity decreased with age after vaccination and was markedly lower in immunosuppressed individuals (OR ~0.16 compared with non-immunosuppressed individuals) [24,25].

Targeted immunosuppressive therapy, particularly anti-CD20 monoclonal antibodies such as rituximab and ocrelizumab used in hematologic malignancies and autoimmune diseases, is strongly associated with impaired vaccine immunogenicity [26]. In the study by Tvito et al., only 1 of 28 Hodgkin lymphoma patients treated with anti-CD20 developed seropositive responses within six months of therapy. Consistent with these findings, Gurion et al. demonstrated that anti-CD20 therapy within the past 12 months was associated with markedly reduced odds of serological response post-vaccination (OR 31.3; 95% CI: 8.4–116.9; *p* < 0.001) [27,28].

Efficacy and safety data on booster doses remain limited, as large-scale clinical trials have often excluded patients with cancer, solid-organ transplants, and rheumatologic disorders, despite these groups representing approximately 3% of the population [29]. Strengths of this study include the use of a clinically validated database, which likely minimized the risk of misclassification bias, and the inclusion of one of the largest reported cohorts of immunosuppressed patients. This large sample size enabled evaluation of the humoral response to a fourth COVID-19 vaccine dose in a real-world setting and in clinically relevant populations that are frequently underrepresented in research.

Our results show that almost 90% of at-risk patients achieved an adequate serological response after the fourth dose. This supports the strategy of administering additional booster doses in immunosuppressed populations, as the vast majority derive measurable immunological benefit. These findings suggest that vaccination policies should continue to prioritize these patients for periodic booster doses. We further recommend risk stratification within the immunosuppressed population, as patients who remain with low titers despite vaccination may benefit from additional protective measures.

Nonetheless, consistent with the inherent limitations of observational research, important variables such as baseline neutralizing antibody titers, unrecognized prior SARS-CoV-2 infection, and cellular immune function were not collected. Inclusion of these parameters could have provided a more comprehensive characterization of the immune response. In our study, immune response was assessed using anti-SARS-CoV-2 IgG binding antibody titers, an accepted marker of immunogenicity, but one that does not capture neutralizing activity or virus-specific cellular immunity (e.g., T-cell responses), both of which play an important role in protection against COVID-19 [20]. Moreover, information on prior SARS-CoV-2 infection was unavailable. Given the high community incidence of Omicron during the study period, undetected asymptomatic infections or hybrid immunity may have influenced antibody titers and potentially led to an overestimation of vaccine-induced responses.

The statistical power for certain subgroup analyses may also have been limited, as small sample sizes and modest effect sizes likely contributed to wide confidence intervals and non-significant results, despite trends consistent with prior evidence. It is possible that the study was underpowered to detect significant differences in the HLI variable, given the limited number of events (inadequate responses) and the broad definition of high-level immunosuppression used, which encompassed conditions with varying degrees of immunological impact.

In evaluating vaccine response in patients with a high degree of immunosuppression, continued research in this area is essential. A better understanding of the factors influencing immune response to vaccination in these patients will help optimize vaccination strategies and guide the development of more effective therapeutic approaches. Furthermore, ongoing monitoring of vaccine response in this population is essential to assess the need for booster doses or additional interventions.

## 5. Conclusions

In conclusion, our results support existing evidence that age and sex influence vaccine response in high-level immunosuppression. However, further studies are required to fully understand the underlying mechanisms and to optimize vaccination strategies for this patient subgroup.

## Figures and Tables

**Figure 1 vaccines-13-00994-f001:**
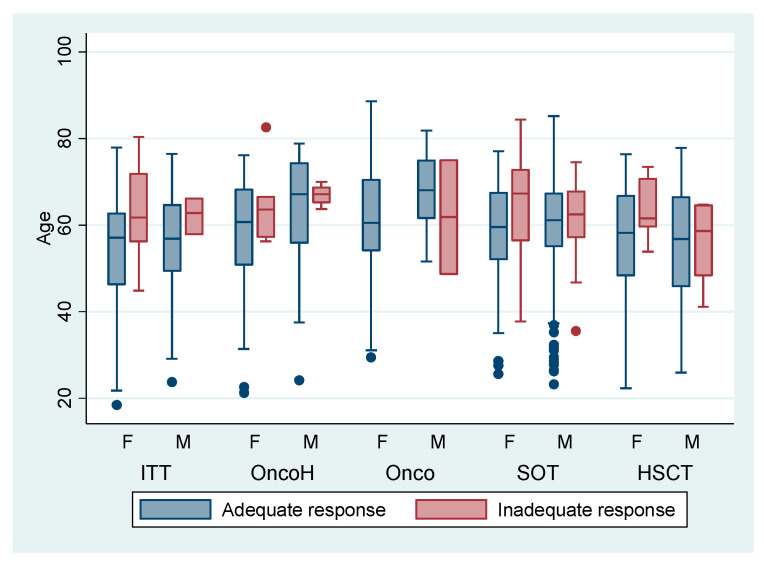
Vaccine response according to age in patients with high levels of immunosuppression. F: female; M: Male; ITT: Immuno-targeted therapy; OncoH: hematologic malignancies; Onco: solid cancer; SOT: solid organ transplant; HSCT: hematopoietic stem cell transplantation. Blue and red dots indicate outliers in the boxplots (observations beyond 1.5 × IQR from the nearest quartile). Dot colors match the category: blue = adequate response; red = inadequate response.

**Figure 2 vaccines-13-00994-f002:**
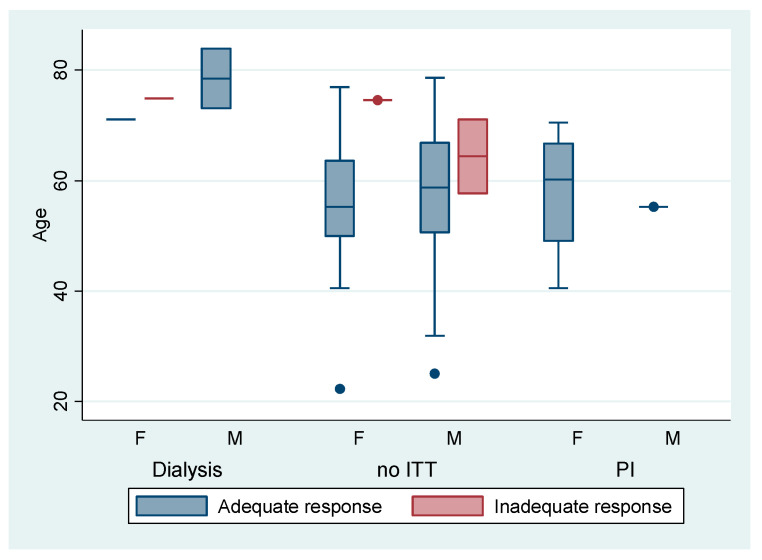
Vaccine response according to age in patients without high-level of immunosuppression. F: female; M: male; noITT: No Immuno-targeted therapy; PI: primary immunosuppression. Blue and red dots indicate outliers in the boxplots (observations beyond 1.5 × IQR from the nearest quartile). Dot colors match the category: blue = adequate response; red = inadequate response.

**Table 1 vaccines-13-00994-t001:** Clinical and epidemiological characteristics of patients vaccinated with a fourth dose of COVID-19 vaccine.

Variable	Adequate HIR Anti-S BAU/mL ≥260 (*n* = 846)	Inadequate HIR Anti-S BAU/mL <260 (*n* = 97)
**Age groups**		
18/39	64 (7.57%)	2 (2.06%)
40/59	347 (41.02%)	30 (30.93%)
60/74	371 (43.85%)	57 (58.76%)
≥75	64 (7.57%)	8 (8.25%)
**Sex**		
Female	323 (323/370)	47 (47/370)
Male	523 (523/573)	50 (50/573)
**Level of immunosuppression**		
Without high-level	58 (6.86%)	4 (4.12%)
High-level	788 (93.14%)	93 (95.88%)
**Clinical conditions**		
SOT ^1^	392 (46.34%)	54 (55.67%)
Immuno-targeted therapy	110 (13.00%)	17 (17.53%)
HSCT ^2^	108 (12.77%)	10 (10.31%)
Hematologic Malignancies	105 (12.41%)	10 (10.31%)
Oncological	73 (8.93%)	2 (2.06%)
Non-immuno-targeted therapy	28 (3.31%)	3 (3.09%)
Dialysis	3 (0.35%)	1 (1.03%)
HIV	22 (2.60%)	0 (0.00%)
Primary immunodeficiencies	5 (0.59%)	0 (0.00%)
**Fourth dose–serology gap** ^3^	18 (RI 16–26)	18 (RI 16–23)

HIR.: humoral immune response. ^1^: Solid organ transplantation; ^2^: Hematopoietic stem cell transplantation; ^3^: interval in days between the 4th dose of the COVID-19 vaccine and the sample collection for post-vaccination serology.

**Table 2 vaccines-13-00994-t002:** Association between vaccine serological response and immunosuppression level.

Variable	Unadjusted OR (95% CI)	*p*	Adjusted OR (95% CI)	*p*
High level of immunosuppression	1.711 (0.612–6.634)	0.303	1.705 (0.600–4.845)	0.316
Age 40–59 years	0.643 (0.411–1.008)	0.054	2.924 (0.680–12.575)	0.149
Age 60–74 years	1.824 (1.193–2.788)	0.005	5.289 (1.255–22.293)	0.023
Age ≥ 75 years	1.098 (0.518–2.329)	0.810	4.236 (0.862–20.809)	0.075
Sex	0.657 (0.421–1.025)	0.049	0.611 (0.399–0.936)	0.024

OR: odds ratio; CI: confidence interval. Reference categories: no high-level immunosuppression, age 18–39 years, and male sex.

## Data Availability

The data are not publicly available due to confidentiality and ethical considerations.

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
