# Peer review of "Serological Response After the Fourth Dose of COVID-19 Vaccine in Highly Immunosuppressed Patients"

_vaccines, 2025, doi:10.3390/vaccines13100994_

Round 1
Reviewer 1 Report
Comments and Suggestions for Authors
The manuscript by Fernández Chávez and colleagues addresses a relevant topic by evaluating the serological response to a fourth dose of the COVID-19 vaccine in immunosuppressed patients, a group with critical public health implications. However, some importante aspects of the study warrant clarification and improvement. First, the abstract displayed in the MDPI submission system is not consistent with that shown in the manuscript file. The authors should better justify the chosen antibody cut-off (260 BAU), as its clinical significance may vary depending on assay and context. Additionally, the confidence interval for the multivariate odds ratio is very wide, requiring a proper discussion to avoid overinterpretation. It would also strengthen the manuscript if the authors elaborated on potential biological or clinical explanations for the observed lower response in women, as this finding contrasts with several reports showing stronger female immunogenicity. Finally, a discussion providing a more in-depth comparison of these results with international data and considering implications for tailored vaccination strategies in highly immunosuppressed populations would add valuable context. Other specific issues are listed below:
Lines 13-14/65-67: The objective does not exactly reflect the title of the manuscript, as the former is not clearly about highly immunosuppressed patients.
Lines 40/56: The unit for "CD4 < 200" was not provided.
Lines 78-80: The definition of inadequate serological response requires revision for clarity; furthermore, the procedure for measurement of serological response was not described.
Line 101: The percentages of female and male who showed and adequate serological response do not correspond to those shown in Table 1.
Lines 108-111/139-140: The data on Table 1 require a more comprehensive description in the main text of the manuscript; moreover, Tables 1 and 2 should use period instead of comma as decimal separator.
Lines 118/127: Figures 1 and 2 lack statistical significance markers.
Lines 220-221: The number of the ethical approval certificate issued by the Institutional Review Board was not provided.
Moderate editing of English language required.
Author Response
We sincerely thank you for the time and effort dedicated to reviewing our manuscript and for your constructive comments, which have helped us to improve the clarity of our work. Please find below our detailed responses to your observations:
- Abstract discrepancy
Thank you for pointing out this inconsistency. We have revised and corrected the abstract to ensure that it matches the manuscript text exactly. - Cutoff of 260 BAU/mL
We agree that this choice required clearer justification. In the Methods section, we have added an explanation supported by the literature to substantiate the use of this cutoff. - Confidence interval interpretation
We acknowledge your comment and have included a specific paragraph in the Discussion section addressing the width of the OR confidence interval in the multivariate analysis: "Although high-level immunosuppression showed a trend toward worse response (OR > 1), the wide confidence interval reflects limited statistical power to confirm an association. This result should therefore be interpreted with caution and does not imply a statistically significant difference." This addition prevents any overinterpretation of the multivariate OR. - Lower response in women
We expanded our analysis of this finding in the Discussion: "Several studies suggest that women, on average, develop higher antibody responses than men after vaccination (18). However, in our study, female patients more frequently had autoimmune diseases requiring intensive immunosuppressive therapy, a factor that may have counteracted the female immunological advantage. (19) This characteristic of our study population provides a plausible biological and clinical explanation for the lower response observed in women, despite the opposite trend described in the general literature." In this way, we contextualize the lower female response as a cohort-specific confounder, not as an intrinsic sex-related difference. - International data and vaccination strategies
Following your recommendation, we made two major additions to the Discussion:
- We incorporated a more extensive comparison with international studies: “In cohorts of kidney transplant recipients and onco-hematological patients, high seroconversion rates were observed after the fourth dose (between 76% and 95% responders), although a subgroup persisted without an adequate humoral response. (23) In the United Kingdom, antibody positivity decreased with age after vaccination and was markedly lower in immunosuppressed individuals (OR ~0.16 compared with non-immunosuppressed individuals). (24) (25)”. These international observations place our findings within the described global trend.
- In another Discussion section, we add: “Our results show that almost 90% of at-risk patients achieved an adequate serological response after the fourth dose. This supports the strategy of administering additional booster doses in immunosuppressed populations, as the vast majority derive measurable immunological benefit. These findings suggest that vaccination policies should continue to prioritize these patients for periodic booster doses. We further recommend risk stratification within the immunosuppressed population, as patients who remain with low titers despite vaccination may benefit from additional protective measures”
- Clarifications and corrections
- We have reformulated the study objective, corrected the unit of measurement for CD4 count, and clarified the definition of “inadequate” serological response, including details on antibody measurement.
- To resolve the confusion in line 101, we clarified sex-specific response rates in the Results section: "Among female patients, 87.3% (323/370) achieved an adequate response, compared with 91.2% (523/573) of male patients"
- In the revised Table 1, we have added footnotes to explain the percentage calculations and have avoided the use of potentially misleading percentages. The number and percentage of adequate responders within each subgroup are now indicated, so that, for example, for females it appears as "323 (87.3% of 370 women)" and for males it appears as "523 (91.2% of 573 men)." This ensures that the table is self-contained and consistent with the text.
- We also expanded the textual description of Table 1 results (lines 108–111/139–140) in response to your comment.
- Formatting and presentation
- Decimal separators have been standardized to international format (period instead of comma).
- Indicators of statistical significance (p<0.05) were added to the figure legends for Figures 1 and 2, improving clarity.
- Ethics approval
We have explicitly added the Ethics Committee approval number in the Methods section:"The study was approved by the Research Ethics Committee of the Ramón y Cajal Hospital (CEI approval code: 07/28/2022 ACTA 438)."
We hope these revisions adequately address your concerns and improve the quality of the manuscript.
Reviewer 2 Report
Comments and Suggestions for Authors
Major findings:
- Impact of Age: The study's finding that older patients had a lower vaccine response is a significant and well-supported contribution, aligning with the established concept of immunosenescence.
- Paradox of Sex-Based Response: The claim that male sex was associated with a higher vaccine response than female sex is a finding that contradicts a large body of established literature on vaccine immunology. This is likely a clinical artifact, possibly due to the disproportionate number of women in the cohort receiving aggressive immunosuppressive therapies for autoimmune diseases.
- Issues with High-Level Immunosuppression (HLI): The study's conclusion that HLI status was not a statistically significant factor for an inadequate response is a highly perplexing finding. This result contradicts extensive evidence that these patients have a diminished capacity to mount a robust immune response to vaccination. This is likely a statistical error stemming from the study's lack of power and the broad definition of the HLI group.
- Methodological Gaps: The manuscript contains critical inconsistencies and omissions, such as conflicting study dates and the use of a single, limited measure of serological response, which provides an incomplete picture of the cohort's true protective immunity
Suggestions:
- Correct Reporting Errors: Resolve the critical inconsistency in the study dates presented in the abstract and methods section.
- Reframe Key Findings: The discussion should be revised to explain that the higher vaccine response observed in males is likely a clinical artifact, and the lack of statistical significance for high-level immunosuppression is probably a Type II statistical error.
- Perform Granular Analysis: Stratify the serological response by specific classes of immunosuppressive therapy to provide a more clinically relevant breakdown of the data.
- Strengthen Limitations: Explicitly discuss the study's limitations, particularly the reliance on a single antibody cutoff and the lack of data on neutralizing antibodies, prior infections, or cellular immunity.
Author Response
We sincerely thank you for your thoughtful review and constructive comments, which have significantly strengthened our manuscript. Below we address each of your observations in detail:
1. Impact of age
Thank you for highlighting this point. We have expanded our discussion on possible mechanisms: “Immunosenescence plays a key role in the response to vaccination and infections in older people and is characterized by age-related immune decline, including decreased antibody production and reduced B- and T-cell activity. (15)(17)”
2. Paradox of sex-based response
You noted that our finding of better response in males contrasts with numerous previous studies. In response, we have rephrased the discussion as follows: “Several studies suggest that women, on average, develop higher antibody responses than men after vaccination (18). However, in our study, female patients more frequently had autoimmune diseases requiring intensive immunosuppressive therapy, a factor that may have counteracted the female immunological advantage. (19) This characteristic of our study population provides a plausible biological and clinical explanation for the lower response observed in women, despite the opposite trend described in the general literature.”
3. High-level immunosuppression (HLI)
We appreciate your concern about unexpecteded results. We have addressed it in the Limitations section:“It is possible that the study was underpowered to detect significant differences in the HLI variable, given the limited number of events (inadequate responses) and the broad definition of high-level immunosuppression used, which encompassed conditions with varying degrees of immunological impact.”
Additionally, we expanded the Discussion section: “Although high-level immunosuppression showed a trend toward worse response (OR > 1), the wide confidence interval reflects limited statistical power to confirm an association. This result should therefore be interpreted with caution and does not imply a statistically significant difference.”
4. Methodological gaps
We are grateful for this constructive criticism, which helped us correct and clarify several aspects:
- Inconsistency in study dates: After revising our records, we corrected the inconsistency. All sections now uniformly state: “the study was conducted between February and August 2022.”
- Assessment of immune response: We explicitly acknowledge that our evaluation was partial. The Limitations section now states: “Consistent with the inherent limitations of observational research, important variables such as baseline neutralizing antibody titers, unrecognized prior SARS-CoV-2 infection, and cellular immune function were not collected. Inclusion of these parameters could have provided a more comprehensive characterization of the immune response. In our study, immune response was assessed using anti-SARS-CoV-2 IgG binding antibody titers, an accepted marker of immunogenicity, but one that does not capture neutralizing activity or virus-specific cellular immunity (e.g., T-cell responses), both of which play an important role in protection against COVID-19. (20) Moreover, information on prior SARS-CoV-2 infection was unavailable. Given the high community incidence of Omicron during the study period, undetected asymptomatic infections or hybrid immunity may have influenced antibody titers and potentially led to an overestimation of vaccine-induced responses.”
We also noted that statistical power for some subgroup analyses was limited, which likely contributed to wide confidence intervals and non-significant results.
5. Granular analysis of immunosuppressive therapy
We greatly value this suggestion. The following details have been added: “In subgroup analyses, solid organ transplant recipients had a slightly lower adequate response rate (≈88%) than the overall cohort, whereas solid cancer patients receiving chemotherapy showed unexpectedly high response rates (≈98% with adequate response; adjusted OR 0.22; p = 0.02, compared with others). Importantly, no single immunosuppressive therapy was significantly associated with humoral nonresponse. (20)”
We acknowledge that some of these analyses are limited by sample size and thus may lack statistical power. Nevertheless, we believe that including them makes the findings more clinically relevant by showing which subgroups of immunosuppressed patients may be at greater risk for poor response.
Finally, we have reinforced the Limitations section to incorporate all of your observations, and we believe that these revisions substantially improve the manuscript by ensuring greater methodological clarity, a more accurate interpretation of the results, and a stronger link between our findings and their clinical relevance.
Reviewer 3 Report
Comments and Suggestions for Authors
This study evaluates serological responses after a fourth dose of mRNA COVID-19 vaccine in 943 high-risk patients in Spain. Overall, 89.7% achieved adequate antibody levels, but weaker responses were observed in older adults and women, with solid organ transplant recipients and those on targeted immunosuppressive therapy showing trends toward lower responses. Age (60-74 years) and sex emerged as independent predictors. The study provides useful real-world data, though its cross-sectional design, lack of prior infection data, and small subgroup sizes limit interpretation.
- In the Introduction, the classification of high-level immunosuppression (HLI) and non-high-level immunosuppression (non-HLI) appears somewhat inconsistent. For example, the manuscript first mentions that patients with Down syndrome, cystic fibrosis, and HIV infection with CD4 < 200 are prioritized for vaccination, but later these groups are categorized under non-HLI. This may cause confusion for the reader.
It is recommended that the authors avoid presenting detailed subgroup classifications too early in the Introduction. Instead, it would be clearer to briefly note that previous studies have applied varying definitions of immunosuppression intensity, and that the present study adopts a specific operational definition, which should be described explicitly in the Methods section. This would reduce ambiguity and improve the logical flow of the Introduction.
- The authors describe this work as an “observational cohort follow-up study.” However, based on the description, the analysis appears to involve patients who received a fourth vaccine dose and had serology measured at a single time point (30–45 days post-vaccination). Without longitudinal follow-up or repeated measurements over time, the design is closer to a cross-sectional analysis rather than a cohort follow-up study.
- The definition of high-level immunosuppression (HLI) is described as being based on the criteria of the Spanish health authorities, but the operational details are not sufficiently specified. It is unclear which specific therapeutic classes (e.g., anti-CD20 agents, mTOR inhibitors), time since transplantation, or HIV-related criteria (such as CD4 thresholds) were used.
- In the Methods section, the authors state that the multivariable analysis adjusted only for age and sex. However, in the Results section, the regression outputs are presented together with univariate comparisons, and it is not always clear to the reader which estimates are derived from bivariate analyses and which come from the multivariable logistic regression.
- The study does not adjust for prior SARS-CoV-2 infection or baseline neutralizing antibodies, which may confound humoral response.
- Table 1 appears to combine both descriptive information and results from univariate logistic regression. It is recommended to present these separately, or alternatively, to combine the univariate and multivariate logistic regression results into a single table to better illustrate the effect of adjustment.
- The Discussion section relies heavily on literature review but provides limited interpretation of the study’s own findings. For instance, the Results indicate a non-significant trend toward poorer response among HLI patients (OR = 1.71, 95% CI: 0.61-6.63), yet the Discussion states broadly that “solid organ transplant recipients exhibited a lower vaccine response,” which may give the impression of statistical significance.
- The Discussion notes that “male patients demonstrated a higher vaccine response compared with female patients,” attributing this partly to the higher prevalence of autoimmune diseases in women. However, the manuscript does not provide sufficient consideration of this study’s own cohort composition—for example, the higher proportion of female patients receiving targeted immunosuppressive therapy—which could act as an important confounder.
- While the Discussion acknowledges limitations such as the absence of neutralizing antibody and cellular immunity data, it does not explicitly highlight the lack of information on prior SARS-CoV-2 infection. Given the widespread circulation of Omicron in Spain during 2022, unrecognized prior infection could have substantially influenced antibody titers and acted as a key confounder.
- The Discussion focuses primarily on immunological mechanisms and literature comparisons but gives limited attention to the clinical or public health significance of the findings. For example, it is not clear whether the results support continued administration of a fourth dose in immunosuppressed patients, or whether they suggest the need for antibody monitoring or prophylactic monoclonal antibody use in certain subgroups. It is recommended that the authors add a brief section highlighting the practical implications of their findings, so that the study provides clearer guidance for clinical decision-making and public health policy.
- Abstract reports OR=5.29, but Results section cites OR=1.824.
Author Response
We are very grateful for your careful review and insightful comments, which have greatly helped us to improve the clarity and consistency of our manuscript. Below we respond point by point:
- Classification of high-level immunosuppression (HLI)
We appreciate this suggestion, as we recognized that the original wording in the Introduction could be confusing. We have restructured the opening paragraph to avoid presenting overly detailed subgroup lists too early. The revised Introduction now reads: “Studies classify recipients of solid organ or hematopoietic stem cell transplants, patients with hematological malignancies, oncology patients receiving chemotherapy, patients on immuno-targeted therapy, and those with combined primary immunodeficiencies as having ‘pathological conditions that induce high-level immunosuppression’.”
In the Methods section, we explicitly describe our definition of HLI. We believe these changes improve clarity and coherence by framing the concept first and then referring the reader to the operational definition later. - Study design
You are correct that the original designation may have been inaccurate. We have corrected the description to reflect the actual data collected. The first sentence of the Methods section now states: “This cross-sectional observational study was conducted between February 15 and August 31, 2022, and included…” - Operational definition of HLI
In response to your comment, we expanded the Methods section to provide a detailed operational definition, explicitly based on Spanish regulations: “High-level immunosuppression was defined according to specific clinical conditions: solid organ or hematopoietic stem cell transplant recipients ≥3 months post-transplant; patients with hematologic malignancies; patients with primary immunodeficiencies classified as severe combined immunodeficiency; cancer patients receiving active cytotoxic chemotherapy; and patients on targeted immunotherapies, specifically anti-CD20 or anti-TNFα agents (e.g., adalimumab, infliximab).” - Univariate vs. multivariate analyses
We agree that our initial presentation could have been clearer. To resolve this, we made the following changes:
- In the Methods section, we now describe our analytical strategy more precisely, indicating that bivariate analyses were first performed, followed by multivariate logistic regression adjusted for age and sex.
- In the Results section, we have separated univariate from multivariate findings, ensuring full transparency.
- Prior SARS-CoV-2 infections and neutralizing antibodies
You are correct that these factors may confound humoral response. We explicitly addressed this limitation in the Discussion: “In our study, immune response was assessed using anti-SARS-CoV-2 IgG binding antibody titers, an accepted marker of immunogenicity, but one that does not capture neutralizing activity or virus-specific cellular immunity (e.g., T-cell responses), both of which play an important role in protection against COVID-19. (20) Moreover, information on prior SARS-CoV-2 infection was unavailable. Given the high community incidence of Omicron during the study period, undetected asymptomatic infections or hybrid immunity may have influenced antibody titers and potentially led to an overestimation of vaccine-induced responses.” - Table 1 presentation
We thank you for pointing this out. Following your suggestion, Table 1 now contains only descriptive characteristics. In Table 2, we present the univariable and multivariable logistic regression analyses side by side, allowing readers to clearly see the effect of adjustment. - Discussion focused on literature review
We acknowledge this important observation. The Discussion has now been revised to reduce excessive reliance on literature and to place greater emphasis on our own results and their interpretation. - Sex-based response and confounding
Your point is very pertinent. We have revised the discussion to explicitly account for cohort-specific confounding: “Several studies suggest that women, on average, develop higher antibody responses than men after vaccination (18). However, in our study, female patients more frequently had autoimmune diseases requiring intensive immunosuppressive therapy, a factor that may have counteracted the female immunological advantage. (19) This characteristic of our study population provides a plausible biological and clinical explanation for the lower response observed in women, despite the opposite trend described in the general literature.”
In short, we now “close the circle” by linking the sex-based difference directly to the structure of our cohort and acknowledging the likelihood of confounding. - Lack of explicit mention of prior infections
We completely agree. We have added this point explicitly in the Discussion:
“Nonetheless, consistent with the inherent limitations of observational research, important variables such as baseline neutralizing antibody titers, unrecognized prior SARS-CoV-2 infection, and cellular immune function were not collected. Inclusion of these parameters could have provided a more comprehensive characterization of the immune response. In our study, immune response was assessed using anti-SARS-CoV-2 IgG binding antibody titers, an accepted marker of immunogenicity, but one that does not capture neutralizing activity or virus-specific cellular immunity (e.g., T-cell responses), both of which play an important role in protection against COVID-19. (20) Moreover, information on prior SARS-CoV-2 infection was unavailable. Given the high community incidence of Omicron during the study period, undetected asymptomatic infections or hybrid immunity may have influenced antibody titers and potentially led to an overestimation of vaccine-induced responses.” - Clinical and public health implications
We fully agree with your recommendation. We have incorporated additional text at the end of the Discussion highlighting the clinical and public health implications of our findings, summarizing their practical relevance for vaccination strategies in immunosuppressed populations. - Discrepancy in reported OR
Thank you for detecting this inconsistency. It arose from including multivariate regression results in the abstract while reporting univariate results elsewhere. We have now clarified the distinction, presenting univariate and multivariate results separately. With this correction, the reported OR values are fully consistent throughout the manuscript and accurately reflect the adjusted findings.
We thank you once again for your thoughtful review and valuable suggestions.
Reviewer 4 Report
Comments and Suggestions for Authors
It is pleasure to review this report. This report discovered the immunological response after fourth dose SARS-CoV-2 vaccine on the immunosuppressed patients. The authors conducted serological analyses across diverse patient cohorts categorized by type, sex, and age, elucidating disparities in post-vaccination immunological responses among these populations. A minor limitation of this study is its sole reliance on serological analysis.
But this result is important for the researcher of vaccine to optimize the schedule of vaccination for special cohorts, and how to improve the vaccine efficacy.
Author Response
We sincerely thank you for your positive and encouraging comments on our work. We appreciate your recognition of the importance of our findings for understanding post-vaccination immune responses in immunosuppressed patients.
We agree with your observation regarding the limitation of relying exclusively on serological analysis. As noted in the revised manuscript, the immune response was assessed using anti-SARS-CoV-2 IgG binding antibody titers, which, although widely accepted as a marker of immunogenicity, does not account for neutralizing activity or virus-specific cellular immunity. We have highlighted this point in the Limitations section to ensure appropriate interpretation of our results.
Once again, thank you for your thoughtful review and for underscoring the potential relevance of our study to vaccine optimization in vulnerable cohorts.
Round 2
Reviewer 1 Report
Comments and Suggestions for Authors
The authors have responded adequately to my comments on the original version by justifying the chosen antibody cut-off, discussing the wide confidence interval for the multivariate odds ratio, elaborating on potential explanations for the observed lower response in women, and considering implications for tailored vaccination strategies in highly immunosuppressed populations, in addition to performing minor adjustments in the manuscript as requested.
Reviewer 3 Report
Comments and Suggestions for Authors
The authors responses to the previous comments well, and the manuscript has been improved a lot. I have no further questions.